# Nationally and regionally representative analysis of 1.65 million children aged under 5 years using a child-based human development index: A multi-country cross-sectional study

**Jan-Walter De Neve**[1]*, **Kenneth Harttgen**[2], **Stéphane Verguet**[3]

1 Heidelberg Institute of Global Health, Medical Faculty and University Hospital, Heidelberg University, Heidelberg, Germany, 2 Department of Humanities, Social and Political Sciences, ETH Zurich, Zurich, Switzerland, 3 Department of Global Health and Population, Harvard T.H. Chan School of Public Health, Boston, Massachusetts, United States of America

* janwalter.deneve@uni-heidelberg.de

**Data Availability Statement:** DHS survey data are available from the DHS Program (https://dhsprogram.com/), and all other study data are

## Abstract

### Background

Education and health are both constituents of human capital that enable people to earn higher wages and enhance people's capabilities. Human capabilities may lead to fulfilling lives by enabling people to achieve a valuable combination of human functionings—i.e., what people are able to do or be as a result of their capabilities. A better understanding of how these different human capabilities are produced together could point to opportunities to help jointly reduce the wide disparities in health and education across populations.

### Methods and findings

We use nationally and regionally representative individual-level data from Demographic and Health Surveys (DHS) for 55 low- and middle-income countries (LMICs) to examine patterns in human capabilities at the national and regional levels, between 2000 and 2017 ($N = 1,657,194$ children under age 5). We graphically analyze human capabilities, separately for each country, and propose a novel child-based Human Development Index (HDI) based on under-five survival, maternal educational attainment, and measures of a child's household wealth. We normalize the range of each component using data on the minimum and maximum values across countries (for national comparisons) or first-level administrative units within countries (for subnational comparisons). The scores that can be generated by the child-based HDI range from 0 to 1.

We find considerable heterogeneity in child health across countries as well as within countries. At the national level, the child-based HDI ranged from 0.140 in Niger (with mean across first-level administrative units = 0.277 and standard deviation [SD] 0.114) to 0.755 in Albania (with mean across first-level administrative units = 0.603 and SD 0.089). There are improvements over time overall between the 2000s and 2010s, although this is not the case

included in the paper and its Supporting Information files.

**Funding:** JWDN was supported by the Alexander von Humboldt Foundation (https://www.humboldt-foundation.de/web/home.html), German Research Foundation (https://www.dfg.de/en/), Baden-Württemberg Ministry of Science, Research and the Arts (https://mwk.baden-wuerttemberg.de/de/startseite/), and Ruprecht-Karls-Universität Heidelberg (https://www.uni-heidelberg.de/en). The funders had no role in study design, data collection and analysis, decision to publish, or preparation of the manuscript.

**Competing interests:** The authors have declared that no competing interests exist.

**Abbreviations:** DHS, Demographic and Health Surveys; FCT, Federal Capital Territory; GDP, gross domestic product; HDI, Human Development Index; LMIC, low- and middle-income country; MICS, Multiple Indicator Cluster Surveys; NCR, National Capital Region; RECORD, REporting of studies Conducted using Observational Routinely collected Data; SD, standard deviation.

for all countries included in our study. In Cambodia, Malawi, and Nigeria, for instance, under-five survival improved over time at most levels of maternal education and wealth. In contrast, in the Philippines, we found relatively few changes in under-five survival across the development spectrum and over time. In these countries, the persistent location of geographical areas of poor child health across both the development spectrum and time may indicate within-country poverty traps.

Limitations of our study include its descriptive nature, lack of information beyond first- and second-level administrative units, and limited generalizability beyond the countries analyzed.

## Conclusions

This study maps patterns and trends in human capabilities and is among the first, to our knowledge, to introduce a child-based HDI at the national and subnational level. Areas of chronic deprivation may indicate within-country poverty traps and require alternative policy approaches to improving child health in low-resource settings.

## Author summary

### Why was this study done?

- Education and health are aspects of human capital that may have important economic benefits for individuals and households. These human capabilities reflect a person's freedom to choose different ways of living and lead a fulfilling life.

- Highlighting how different human capabilities are produced together can support efforts to synergistically reduce the wide disparities in health and education across populations.

- Few recent efforts, however, have been made to expand measurement to include education, health, and wealth—and, to our knowledge, none have looked at indicators that are focused on improving child health at the national and subnational level.

### What did the researchers do and find?

- We used nationally and regionally representative individual-level data from 1.65 million under-five children in 55 low- and middle-income countries (LMICs) to examine patterns in human capabilities at the national and regional levels, between 2000 and 2017.

- We graphically analyzed human capabilities, separately for each country, and proposed a novel child-based Human Development Index (HDI) based on under-five survival, maternal educational attainment, and measures of a child's household wealth.

- We found considerable heterogeneity in the child-based HDI across countries—ranging from 0.140 in Niger to 0.755 in Albania—as well as within countries.

- We found improvements over time overall although this is not the case for all countries. The persistent location of geographical areas of poor child health across both the development spectrum and time may indicate within-country poverty traps.

**What do these findings mean?**

- Areas of chronic deprivation across and within countries may require alternative policy approaches to improving human capabilities jointly in low-resource settings.

- These findings may point decision-makers working towards achieving the Sustainable Development Goals to more targeted efforts to further reduce persistent health disparities.

- Our approach to compute a capability index is straightforward to implement and could be easily adapted to other indicators, populations, and settings. Limitations of our study include its descriptive nature, lack of information beyond first- and second-level administrative units, and limited generalizability beyond the countries analyzed.

## Introduction

Education and health are both aspects of human capital that enable people to earn higher wages [1] and enhance people's capabilities to lead fulfilling lives. A better understanding of how these different capabilities are produced together could point to opportunities to help synergistically reduce the wide disparities in health and education across populations [2–7]. In 1990, a Human Development Index (HDI) was developed as an alternative to the traditional unidimensional economic measure of development (e.g., the gross domestic product [GDP] [8]) and was initially calculated as the arithmetic mean of normalized values of life expectancy, educational attainment, and income [9]. More recently, in 2010, the geometric mean was introduced to compute the HDI, which reduced the level of substitutability between dimensions (i.e., a low achievement in one dimension could no longer be linearly compensated for by a high achievement in another dimension). Nevertheless, while the current HDI sets out to measure the development of 'members of a society,' it does not take into account the full distribution and co-distribution of the different human capabilities within a country. The HDI has largely remained a national aggregate index, rather than a measure of development at the subnational or household level, and does not fully encompass within-country distributions. Furthermore, the HDI considers life expectancy, as opposed to other measures of health that may be more sensitive to socioeconomic inequalities.

A few studies have aimed to calculate an HDI for subnational units—including by within-country income groups [10,11], and internal migration status [12]—using household-level data for multiple low- and middle-income countries (LMICs) [13–16]. One challenge with calculating health or mortality outcomes at the subnational or household level is that there may be limitations due to small sample size and limited variation without a continuous variable (i.e., in most households either none, one, or two members died, resulting in household mortality rates clustered around 0 or values such as 0.50). To address this issue, a handful of studies have calculated mortality rates using imputation methods. One study, for instance, used Demographic and Health Surveys (DHS) data to impute child mortality by regressing child mortality on household and community socioeconomic characteristics, applying life table systems [17] to estimate household-specific life expectancy at birth and subsequently calculating a health index using estimated life expectancy for each household [13].

In this study, we aim to make 2 contributions to the literature on human development and capabilities, taking the illustrative example of under-five mortality. First, we used nationally

and regionally representative individual-level data from 55 LMICs to show current patterns as well as trends in human capabilities at the national and subnational level. Second, we used under-five mortality, maternal educational attainment, and household wealth as our measures of health, education, and wealth, respectively, to introduce a novel child-based capability index (i.e., a child-based version of the HDI). The premise of the study was not to replace existing measures of human development but rather to explore measures that are particularly sensitive to population-level social and economic inequalities and that are likely to be highly policy relevant [18–20]. In doing so, this study aimed to compute a child-based capability index that is straightforward to implement as a summary metric for decision-makers seeking to bolster human capabilities in the post-2015 development era and could be easily adapted to other indicators, populations, and settings.

## Methods

We developed the case study of a child-based capability index, and in this section, we present the main steps underlying its construction, which we then illustrate by the application to a selection of LMICs for which household survey data were available. There was a written prospective protocol for the study (S1 File), which was adapted in response to peer review comments to further clarify our methodological approach and results. This study is reported as per the REporting of studies Conducted using Observational Routinely-collected Data (RECORD) guideline (S1 Checklist).

### Data sources and sample population

To illustrate the computation of our child-based capability index, we extracted data on age, sex, geographical location of household, under-five survival, maternal educational attainment, and household wealth from 2 DHS surveys (one carried out during the 2010s and, when available, one carried out during the 2000s) for 55 LMICs. A key advantage of the DHS is the availability of comparable data for multiple countries and consistent quality of reporting and data over time [21]. The country and survey selections were chosen to be illustrative rather than exhaustive: our aim was to include a mix of LMICs for which nationally and regionally representative data on our outcomes were available (our approach could easily be reproduced to a larger number of countries and years). The countries included in our analysis are listed in Table 1. For each survey, the DHS birth recode files provide a full birth history of all women interviewed during the survey and include data for the mother of each child [21]. In DHS surveys, the birth history data are typically collected from all women ages 15 to 49 years. We included all children born in a household surveyed by the DHS and for whom complete data on under-five survival status, maternal education, and household assets (wealth) were available. In our main analysis, we included under-five survival data for all children born in the past 10 years preceding the survey. We considered alternative sample specifications in sensitivity analyses described below.

### Underlying outcome variables: Health, education, and wealth

We built on the HDI approach, therefore we assembled 3 similar components towards construction of a child-based capability index. First, we selected maternal educational attainment, calculated as the highest grade or level of formal schooling attained (years of schooling) by the mother of the child ('Edu'). We used maternal education as opposed to average household or paternal education, since maternal education is a strong marker of child health, reflects gender disparities across households in accessing formal education, and is commonly available across surveys in LMICs [18,22,23]. Second, we selected a child's household wealth index, ranging

**Table 1. Selected characteristics of study countries.**

| Country | Survey year | Under-five children (number) | Under-five mortality (per 1,000) | Maternal schooling (mean number of years) |
|---|---|---|---|---|
| *Country* | | | | |
| Afghanistan | 2015–2016 | 66,306 | 57 | 1.0 |
| Albania | 2017–2018 | 5,811 | 6 | 12.0 |
| Angola | 2015–2016 | 25,598 | 67 | 4.2 |
| Armenia | 2015–2016 | 3,515 | 10 | 11.9 |
| Bangladesh | 2014 | 16,792 | 51 | 5.3 |
| Benin | 2017–2018 | 25,343 | 88 | 1.8 |
| Burkina Faso | 2010 | 29,644 | 125 | 0.8 |
| Burundi | 2016–2017 | 25,495 | 71 | 2.7 |
| Cambodia | 2014 | 14,616 | 44 | 4.5 |
| Cameroon | 2011 | 22,195 | 110 | 4.7 |
| Chad | 2014–2015 | 37,925 | 127 | 1.6 |
| Colombia | 2015 | 24,407 | 17 | 9.4 |
| Comoros | 2012 | 6,091 | 47 | 3.7 |
| Congo, Dem. Rep. | 2013–2014 | 34,290 | 95 | 5.1 |
| Congo, Rep. | 2011–2012 | 17,221 | 66 | 7.1 |
| Côte d'Ivoire | 2011–2012 | 14,903 | 102 | 2.0 |
| Dominican Republic | 2013 | 7,184 | 33 | 9.8 |
| Egypt, Arab Rep. | 2014 | 29,661 | 29 | 8.3 |
| Ethiopia | 2016 | 21,606 | 73 | 1.5 |
| Gabon | 2012 | 11,182 | 56 | 7.5 |
| Gambia, The | 2013 | 14,983 | 52 | 2.9 |
| Ghana | 2014 | 11,430 | 61 | 5.6 |
| Guatemala | 2014–2015 | 24,263 | 36 | 4.6 |
| Guinea | 2012 | 14,081 | 115 | 1.2 |
| Haiti | 2016–2017 | 13,144 | 74 | 5.3 |
| Honduras | 2011–2012 | 21,070 | 31 | 4.5 |
| India | 2015–2016 | 536,386 | 49 | 5.7 |
| Indonesia | 2012 | 36,714 | 39 | 8.8 |
| Jordan | 2012 | 20,346 | 20 | 11.1 |
| Kenya | 2014 | 42,847 | 50 | 7.4 |
| Kyrgyz Republic | 2012 | 7,685 | 30 | 12.1 |
| Lesotho | 2014 | 6,026 | 84 | 7.6 |
| Liberia | 2013 | 15,515 | 97 | 3.4 |
| Malawi | 2015–2016 | 34,598 | 65 | 5.2 |
| Maldives | 2016–2017 | 6,319 | 19 | 9.6 |
| Mali | 2012–2013 | 19,863 | 91 | 1.0 |
| Mozambique | 2011 | 20,640 | 93 | 2.8 |
| Namibia | 2013 | 9,433 | 52 | 8.3 |
| Nepal | 2016 | 10,402 | 43 | 4.0 |
| Niger | 2012 | 25,117 | 124 | 0.7 |
| Nigeria | 2013 | 61,629 | 121 | 4.4 |
| Pakistan | 2017–2018 | 25,677 | 73 | 3.9 |
| Peru | 2012 | 19,600 | 22 | 8.7 |
| Philippines | 2017 | 22,158 | 26 | 10.2 |
| Rwanda | 2014–2015 | 15,876 | 55 | 4.2 |
| Senegal | 2017 | 23,895 | 52 | 2.2 |

(*Continued*)

**Table 1.** (Continued)

| Country | Survey year | Under-five children (number) | Under-five mortality (per 1,000) | Maternal schooling (mean number of years) |
|---|---|---|---|---|
| Sierra Leone | 2013 | 24,348 | 156 | 1.8 |
| South Africa | 2016 | 6,994 | 48 | 10.4 |
| Tajikistan | 2017 | 11,190 | 30 | 10.0 |
| Tanzania | 2015–2016 | 19,260 | 67 | 5.3 |
| Timor-Leste | 2016 | 14,387 | 37 | 6.7 |
| Togo | 2013–2014 | 13,931 | 81 | 3.1 |
| Uganda | 2016 | 30,086 | 64 | 5.7 |
| Zambia | 2013–2014 | 26,180 | 69 | 5.8 |
| Zimbabwe | 2015 | 11,314 | 74 | 8.9 |
| *Global average* | - | 30,000 | 63 | 5.6 |

Table 1 shows study countries and most recent DHS survey years included, as well as under-five mortality (per 1,000 live births) and maternal schooling (years) at the national level for each survey. Under-five mortality was calculated using a binary variable indicating whether the child was alive or not at age 5 years at the time of the survey using data on all children born in the past 10 years in a household surveyed by the DHS and for whom complete data on survival status, maternal education, and household wealth were available. Additional details on the construction of outcomes and sensitivity analyses are presented in the main text and S1 Text and S2 Text. Survey year indicates the year(s) in which data collection for the survey was carried out. Survey sample weights were used as provided by the DHS.

**Abbreviation:** DHS, Demographic and Health Surveys

from 1 to 5 (5 being the wealthiest) ('Wealth'). Wealth quintile is a measure of household wealth relative to other households across countries (for national comparisons) or within-countries (for subnational comparisons) [24,25] and is based on ownership of household assets and quality of the dwelling [26]. The same household wealth quintile was assigned to all children living in the household surveyed by the DHS, and maternal educational attainment was assigned to all children from her birth history. Third, we created a binary variable indicating whether the child was alive or not at age 5 years at the time of the survey ('U5S'): under-five survival would mean a higher score on all 3 components would be considered positive (S1 Fig). We avoided the use of imputation in generating all 3 outcome variables. Additional details on outcomes and sensitivity analyses are provided in S1 Text and S2 Text.

## Analysis

Our analysis proceeded in 4 steps. First, under-five survival rates at the subnational level were computed. To increase the number of observations in each (administrative unit) cell, we calculated under-five survival, maternal educational attainment, and household wealth by the first-level administrative unit available in each DHS. The first-level administrative units considered include, for instance, regions (the Philippines) and provinces (Afghanistan) as well as states and union territories (India). The aggregate estimate for each component was derived from individual-level data by averaging over individuals in a given subnational administrative unit ($\overline{Survival}_{c,u}$, $\overline{Education}_{c,u}$, and $\overline{Wealth}_{c,u}$) in which $\overline{Survival}_{c,u}$ represented the proportion of children in country-year $c$ and administrative unit $u$ who survived and $\overline{Education}_{c,u}$ and $\overline{Wealth}_{c,u}$ represented the average maternal educational attainment and wealth quintile in that group. For most DHS surveys, averages at these subnational levels provided regionally representative estimates. We used DHS survey sample weights to generate representative samples. Average sample size per country in our study was 30,000 children across 12 first-level administrative units (Table 1).

Second, we graphically analyzed under-five survival, education, and wealth. To do so, we plotted contour maps (akin to heat maps), in which we displayed $z$ (under-five survival) as filled contours in ($x$ = wealth, $y$ = education). We show three-dimensional data where under-five survival of the sample was represented by the color so that points with equal under-five survival in the graph have the same color. For each $z$ value of under-five survival, we had a position for the 2 other $x$ and $y$ components of wealth and maternal education, respectively. Figures were generated using the 'twoway contour' command in Stata MP 15.1 (College Station, TX), which displays $z$ as filled contours in ($x$, $y$), using the default thin-plate-spline interpolation method [27]. To increase the resolution of the heat maps, we set the range in levels of under-five survival to 10. For visualization purposes, we normalized the range of each of the 3 components (rescaled from 0 to 1) as follows:

$$\text{Normalized Indicator}_{c,u} = \frac{\text{Indicator}_{c,u} - \text{Indicator}_{c,u}^{\min}}{\text{Indicator}_{c,u}^{\max} - \text{Indicator}_{c,u}^{\min}} \tag{1}$$

To examine time trends in under-five survival by maternal educational attainment and wealth, we show results for selected countries with first- or second-level administrative boundaries that have remained largely consistent over the study period (2000s to 2010s). To further increase the resolution of the heat maps, we used the lowest available administrative unit that is consistently available across the DHS country surveys (either first- or second-level administrative units). As an example, we show child health along the development spectrum for the 36 second-level administrative units (states) and the Federal Capital Territory (FCT) of Nigeria—rather than aggregating by first-level administrative units—in 2003 and in 2013. This approach allowed us to examine subnational shifts in child health for selected countries across the development spectrum and over time during the final run-up towards the Millennium Development Goals.

Third, we calculated a summary metric for the child-based capability index, as the geometric mean of the 3 normalized components (see Eq 2 below). While the geometric mean has been commonly used to summarize aggregate measures of human development, it has been applied infrequently to individual-level data from population-based surveys [13]. We computed the child-based capability index at the national level for all 55 countries $c$ and first-level administrative units $u$. When calculating the child-based capability index at the national level, we normalized the range of each component (rescaled from 0 to 1) using data on the minimum and maximum values across countries (akin to Eq 1). When calculating the child-based capability index at the regional level for all countries, we used first-level administrative units (as opposed to second-level administrative units) because (i) data on first-level administrative units were available for all countries in the DHS, whereas this was not the case for lower-level administrative units; (ii) for most DHS surveys, averages at these subnational levels provided regionally representative estimates; and (iii) the components of our index can be estimated precisely using a large number of observations for first-level administrative units. When calculating the index at the regional level, we also estimated the mean across first-level administrative units $u$ in each country and the corresponding standard deviations (SDs) to provide an estimate of within-country variation:

$$\text{Capability Index}_{c,u} = \sqrt[3]{U5S_{normalized,c,u} * Edu_{normalized,c,u} * Wealth_{normalized,c,u}}. \tag{2}$$

Fourth, showing results for geographical regions may improve the usability and interpretation of our approach and may point decision-makers to more targeted efforts to increase child-based capabilities in at-risk regions. We therefore plotted the child-based capability index, as calculated in Eq 2, for geographical regions. Since a geographical analysis for all 55

countries would be a relatively large undertaking, we show results for a selected country. To illustrate within-country variation, we used individual-level child data from the DHS of the Philippines of 2017 ($N$ = 22,158 children). After constructing the child-based capability index for each geographical region of the Philippines, we grouped regions in 10 groups of similar size (deciles) based on their index values. We then mapped the index for each of the deciles using a base map with geographic boundaries of the Philippines provided by Natural Earth (https://www.naturalearthdata.com/).

## Sensitivity analyses

We conducted a range of sensitivity analyses to test the robustness of our findings. First, child survival is reported through birth histories of mothers in the DHS, which may be affected by recall bias. We therefore included alternative specifications including all children born within the past 5 years of the survey (instead of children born within the past 10 years), as well as the full birth history of children. Second, we used alternative definitions for the index components. We used infant survival (defined as survival within the first year of birth) as opposed to under-five survival. We also restricted the sample to children of mothers aged 15 to 30 years because recent changes in the education sector would be reflected in younger rather than older women cohorts. Third, we used the arithmetic mean (as opposed to the geometric mean) to calculate the child-based capability index [28]. Fourth, in our main approach, the normalization of our component values into a 0-to-1 range was done relative to the minimum and maximum values across countries (for national comparisons) or administrative units within countries (for subnational comparisons). We constructed a subindex for each component based on alternative 'goalpost' (reference) values. For instance, we used maternal educational attainment of 15 years as a reference value [29]. Fifth, in our main analysis, we constructed a measure of wealth that allowed comparisons of the child-based capability index across countries [26,30]. As a sensitivity analysis, we present results using the DHS-provided wealth index [21]. The wealth index built into the DHS takes into account country-specific differences, and this approach can be extended in a straightforward manner by others seeking to replicate our work. Sixth, we compared our child-based capability index with other commonly used indices at both the national and subnational level. At the national level, for instance, we compared our child-based capability index to the HDI, the World Bank's Human Capital Index [31], and the Socio-Demographic Index developed by the Global Burden of Disease study [32]. Additional details on sensitivity analyses are provided in S2 Text.

## Data and ethics

This was a complete case analysis, and all analyses were conducted in Stata MP 15.1 (College Station, TX). DHS survey data are available from the DHS Program (https://dhsprogram.com/), and all other study data are included in the paper and its supporting files. This study was preregistered and approved by the Heidelberg University Hospital Ethics Committee (S-271/2019). DHS survey protocols were also approved by country-specific Institutional Review Boards.

## Results

In Table 1, we show average under-five mortality and maternal educational attainment at the national level for the most recent survey for all 55 countries. Under-five mortality ranged from 6 reported deaths per 1,000 live births in Albania to 156 reported deaths per 1,000 live births in Sierra Leone. Maternal educational attainment ranged from 0.7 years (average) in Niger to 12.1 years in Kyrgyz Republic.

In S1 Table, we show under-five mortality, maternal education, and household wealth for each first-level administrative unit for the countries included in our study (e.g., provinces, regions, and states and union territories). We find large heterogeneity not only across countries but also within countries. Average under-five mortality in Afghanistan ranged from 4 reported deaths per 1,000 live births in Helmand province to 150 reported deaths per 1,000 live births in Nuristan province. Similarly, in India, average maternal educational attainment ranged from 3.0 years of schooling in Bihar to 11.8 years of schooling in Kerala.

In Fig 1, we display normalized under-five survival, maternal education, and household wealth ranging from 0 (worst) to 1 (best) for selected countries. The different components were calculated at the subnational level. The color in the figure indicates the level of under-five survival. We show results separately by country as well as over time between the 2000s and 2010s to examine progress during the study period using the lowest available administrative units that have remained consistent over the study period (either first or second level). We

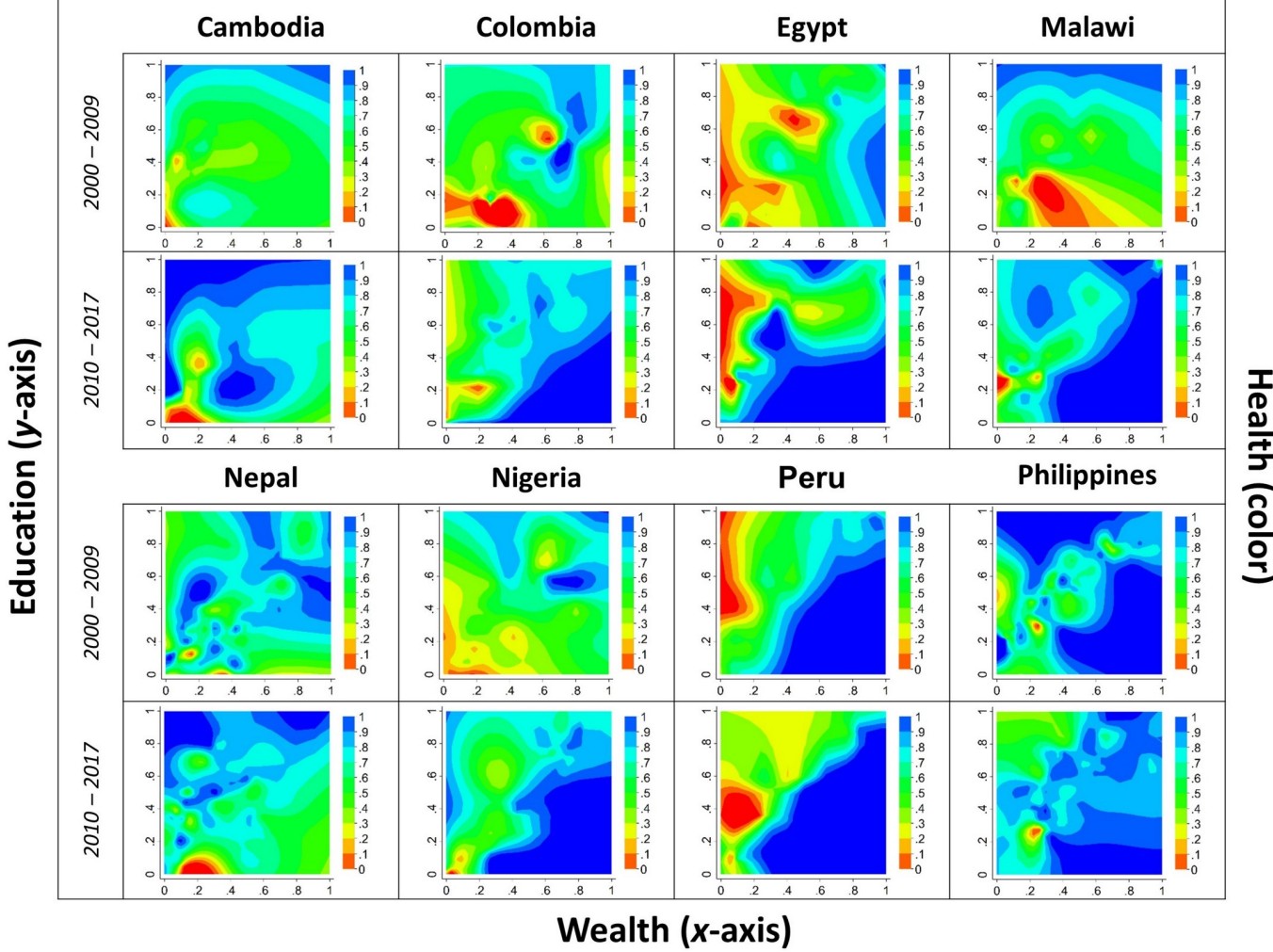

**Fig 1. Child-based capabilities in LMICs, 2000–2017.** Fig 1 shows 3 axes, including health (under-five survival), wealth (household wealth), and education (maternal education), using 2 DHS surveys (2000s and 2010s), separately for each country. Under-five survival is represented by the color (blue being high survival and red being low survival) so that points with equal under-five survival in the graph have the same color. For each $z$ value of under-five survival, there is a position for the 2 other $x$ and $y$ components of wealth and education, respectively. The range of the 3 components was normalized (rescaled from 0 to 1) using data on the minimum and maximum values across administrative units within countries. Additional details on the components and sensitivity analyses are presented in the main text and S1 Text and S2 Text. DHS, Demographic and Health Surveys; LMIC, low- and middle-income country.

find substantial heterogeneity in progress in improving child health over time. In Nigeria, for instance, we find relatively strong progress in child health across the development spectrum between the 2000s and 2010s. We also find large heterogeneity in the location of 'red zones' of low under-five survival within a country. In Malawi in the 2000s, for instance, poor child health (red zones in the figure) was concentrated in areas with relatively high on average levels of household wealth. In contrast, by the 2010s, low under-five survival was largely concentrated in areas with on average poorer households.

In Table 2 (column 3), we show results for our calculations for a child-based capability index at the national level, where countries are ranked by their score on the index. We find that Albania had the highest index in our analysis (0.755; with mean across first-level administrative units = 0.603; SD 0.089), while Niger had the lowest index (0.140; with mean across first-level administrative units = 0.277; SD 0.114). The average score across all 55 countries in our study was 0.466.

In Table 2 (column 6) and S1 Table (column 6), we show within-country variations in the child-based capability index. We identified large variation across states and union territories in India, for instance, where the index ranged from 0.294 in Bihar to 0.783 in Kerala (mean = 0.557; SD 0.119). Conversely, we identified relatively little variation across first-level administrative units in Bangladesh, the Dominican Republic, and Nepal. In Bangladesh, the index ranged from 0.425 in Sylhet Division to 0.547 in Chittagong Division (mean = 0.501; SD 0.045). In Fig 2, we map results for the child-based capability index at the subnational level for geographical regions. We display results for the Philippines, as an illustrative example, where the child-based capability index ranged from 0.323 in the Autonomous Region in Muslim Mindanao (shown in dark red on the map) to 0.760 in the National Capital Region (NCR) (shown in dark blue).

Our results were generally consistent across sensitivity analyses, including when using alternative outcomes, goalpost values, and sample specifications (S2 Fig, S3 Fig, and S2 Table). Potential concerns such as limited sample size, recall bias among mothers, and our methodological approach to calculate under-five mortality are unlikely to substantially affect our main findings. In S3 Table and S4 Table, we present results for side-by-side comparisons with other indices. We find that the correlation with our child-based capability index was highest for the Socio-Demographic Index and lowest for the Human Capital Index (Pearson's correlation coefficients = 0.96 and 0.84, respectively; $p$-values for tests of independence < 0.01).

## Discussion

Using nationally and regionally representative data from 1,657,194 children, this retrospective analysis makes a number of contributions to our understanding of where human capabilities are produced jointly [33]. First, we find substantial heterogeneity in child health across countries as well as within countries and over time. At the national level, the child-based capability index was highest in Albania and lowest in Niger. At the subnational level, geographical areas of low under-five survival existed in expected areas—i.e., areas with relatively low levels of average maternal educational attainment and household wealth—as well as in unexpected areas along the development spectrum (displayed as 'red zones' in Fig 1). Second, our study shows trends in child health over time. We find improvements over time overall between 2000 and 2017, although this is not the case for all countries included in our study. In Cambodia and Nigeria, for instance, under-five survival improved in geographical areas at most levels of average maternal educational attainment and household wealth, whereas in Peru and the Philippines, for instance, under-five survival was distributed relatively consistently over time. Third, our analysis reveals changes in the location of areas of low under-five survival both along the development spectrum within countries and over time. In Egypt and Malawi, for

**Table 2. Countries ranked by child-based capability index.**

| Country | Survey year | National child-based capability index | First-level administrative units | | | |
|---|---|---|---|---|---|---|
| | | | Units | n_DHS | Mean | SD |
| *Country* | | | | | | |
| Albania | 2017–2018 | 0.755 | Counties | 12 | 0.603 | 0.089 |
| Jordan | 2012 | 0.739 | Regions | 3 | 0.625 | 0.021 |
| Maldives | 2016–2017 | 0.715 | Provinces | 6 | 0.572 | 0.111 |
| Dominican Republic | 2013 | 0.700 | Regions, capital | 9 | 0.520 | 0.061 |
| Colombia | 2015 | 0.682 | Regions | 6 | 0.596 | 0.099 |
| Armenia | 2015–2016 | 0.681 | Divisions | 11 | 0.619 | 0.083 |
| Philippines | 2017 | 0.665 | Regions | 17 | 0.553 | 0.102 |
| Kyrgyz Republic | 2012 | 0.663 | Regions, capital | 9 | 0.646 | 0.111 |
| South Africa | 2016 | 0.655 | Provinces | 9 | 0.640 | 0.070 |
| Egypt, Arab Rep. | 2014 | 0.651 | Regions | 6 | 0.621 | 0.135 |
| Indonesia | 2012 | 0.644 | Provinces | 33 | 0.557 | 0.106 |
| Peru | 2012 | 0.603 | Regions, capital | 25 | 0.554 | 0.132 |
| Tajikistan | 2017 | 0.589 | Regions, capital | 5 | 0.622 | 0.109 |
| Gabon | 2012 | 0.561 | Provinces, cities | 10 | 0.481 | 0.104 |
| Namibia | 2013 | 0.516 | Regions | 13 | 0.539 | 0.097 |
| Honduras | 2011–2012 | 0.515 | Departments | 18 | 0.466 | 0.101 |
| Timor-Leste | 2016 | 0.500 | Municipalities | 13 | 0.491 | 0.092 |
| Zimbabwe | 2015 | 0.496 | Provinces, cities | 10 | 0.566 | 0.107 |
| Ghana | 2014 | 0.481 | Regions | 10 | 0.451 | 0.172 |
| Guatemala | 2014–2015 | 0.467 | Regions | 8 | 0.457 | 0.110 |
| India | 2015–2016 | 0.467 | States, union territories | 36 | 0.557 | 0.119 |
| Cambodia | 2014 | 0.449 | Regions, capital | 19 | 0.444 | 0.092 |
| Congo, Rep. | 2011–2012 | 0.448 | Departments | 12 | 0.425 | 0.129 |
| Lesotho | 2014 | 0.441 | Districts | 10 | 0.534 | 0.091 |
| Angola | 2015–2016 | 0.437 | Provinces | 18 | 0.374 | 0.086 |
| Pakistan | 2017–2018 | 0.428 | States, territories, capital | 6 | 0.419 | 0.189 |
| Kenya | 2014 | 0.415 | Provinces | 8 | 0.520 | 0.168 |
| Nepal | 2016 | 0.394 | Provinces | 7 | 0.520 | 0.116 |
| Comoros | 2012 | 0.390 | Islands | 3 | 0.399 | 0.099 |
| Nigeria | 2013 | 0.378 | Zones | 6 | 0.498 | 0.180 |
| Haiti | 2016–2017 | 0.373 | Departments, capital | 11 | 0.486 | 0.080 |
| Zambia | 2013–2014 | 0.368 | Provinces | 10 | 0.471 | 0.110 |
| Togo | 2013–2014 | 0.367 | Regions, capital | 6 | 0.398 | 0.139 |
| Gambia, The | 2013 | 0.365 | Local government areas | 8 | 0.370 | 0.153 |
| Senegal | 2017 | 0.355 | Regions | 14 | 0.302 | 0.112 |
| Bangladesh | 2014 | 0.354 | Divisions | 7 | 0.501 | 0.045 |
| Cameroon | 2011 | 0.347 | Regions, capital | 12 | 0.529 | 0.169 |
| Tanzania | 2015–2016 | 0.335 | Regions | 30 | 0.524 | 0.097 |
| Uganda | 2016 | 0.326 | Regions | 15 | 0.464 | 0.137 |
| Côte d'Ivoire | 2011–2012 | 0.316 | Districts | 11 | 0.321 | 0.097 |
| Liberia | 2013 | 0.308 | Regions, subregions | 5 | 0.372 | 0.107 |
| Benin | 2017–2018 | 0.288 | Departments | 12 | 0.371 | 0.116 |
| Congo, Dem. Rep. | 2013–2014 | 0.271 | Provinces | 11 | 0.498 | 0.102 |
| Malawi | 2015–2016 | 0.268 | Regions | 3 | 0.501 | 0.057 |
| Rwanda | 2014–2015 | 0.267 | Provinces | 5 | 0.468 | 0.096 |

(*Continued*)

**Table 2.** (Continued)

| Country | Survey year | National child-based capability index | First-level administrative units | | | |
|---------|-------------|---------------------------------------|------|-------|------|------|
| | | | Units | n_DHS | Mean | SD |
| Afghanistan | 2015–2016 | 0.262 | Provinces | 34 | 0.260 | 0.091 |
| Mozambique | 2011 | 0.260 | Provinces, capital | 11 | 0.435 | 0.131 |
| Guinea | 2012 | 0.222 | Regions | 8 | 0.280 | 0.126 |
| Sierra Leone | 2013 | 0.205 | Provinces | 4 | 0.364 | 0.139 |
| Mali | 2012–2013 | 0.193 | Regions, capital | 6 | 0.295 | 0.132 |
| Burundi | 2016–2017 | 0.191 | Provinces | 18 | 0.411 | 0.096 |
| Burkina Faso | 2010 | 0.174 | Regions | 13 | 0.251 | 0.101 |
| Ethiopia | 2016 | 0.159 | Regions, chartered cities | 11 | 0.345 | 0.152 |
| Chad | 2014–2015 | 0.158 | Regions | 21 | 0.269 | 0.109 |
| Niger | 2012 | 0.140 | Regions, capital | 8 | 0.277 | 0.114 |
| *Global average* | - | 0.425 | Units | 12 | 0.466 | 0.111 |

Table 2 shows the child-based capability index at the national level (column 3) and corresponding within-country variation (columns 6 and 7). The index was calculated using the geometric mean of under-five survival (1 minus under-five mortality), maternal schooling (years), and household wealth index (quintiles). Each of the 3 components of the national-level child-based capability index was based on data from the entire study population (column 3). The range of each component was normalized (rescaled from 0 to 1) using data on the minimum and maximum values across countries (for national comparisons) or first-level administrative units within countries (for subnational comparisons). DHS surveys are typically representative at the regional level or groups of regions. Survey year indicates the year(s) in which data collection for the survey was carried out.

**Abbreviations:** DHS, Demographic and Health Surveys; n_DHS, number of first-level administrative units available in the DHS; SD, standard deviation

instance, low under-five survival shifted from areas with on average wealthier households to areas with on average poorer households. In contrast, in the Philippines, the consistent location of areas of poor child health across both the development spectrum and over time may indicate areas of chronic deprivation among populations at risk. These areas may indicate within-country poverty traps and require alternative policy approaches to improving child health.

While child health was generally better in wealthier areas, we identified a number of areas of poor child health with relatively high average levels of human capital. In Egypt and Peru, for instance, low under-five survival was observed in areas with high levels of educational attainment (upper left corner in Fig 1), in particular for girls under five (see S4 Fig for results disaggregated by a child's sex in Egypt and Peru) [34]. Although this finding appears counter-intuitive, a growing literature suggests mixed child health returns to additional years of maternal schooling [35–38]. A recent systematic review examined evidence for a causal link between maternal education and child health and found that parental schooling may play a more muted role in parents' decisions about whether and how much to invest in their children's health than previously suggested [39]. Moreover, even well-educated parents seeking to correct common health risks in their children may lack access to high quality primary healthcare services or face high out-of-pocket expenditures [40]. There may also be threshold levels to see an effect of parental schooling on child health outcomes (e.g., primary schooling alone may not be enough to see a protective effect on child health). One reason for threshold levels may be overcrowding or poor quality of instruction at lower school levels [41].

One challenge with calculating child mortality outcomes, however, is that there is limited variation at the household level (since in most households either none, one, or two children died, resulting in household mortality rates clustered around 0 or values such as 0.50). Prior studies have therefore regressed child mortality on household and community socioeconomic characteristics, applied life table systems to estimate household-specific life expectancy at

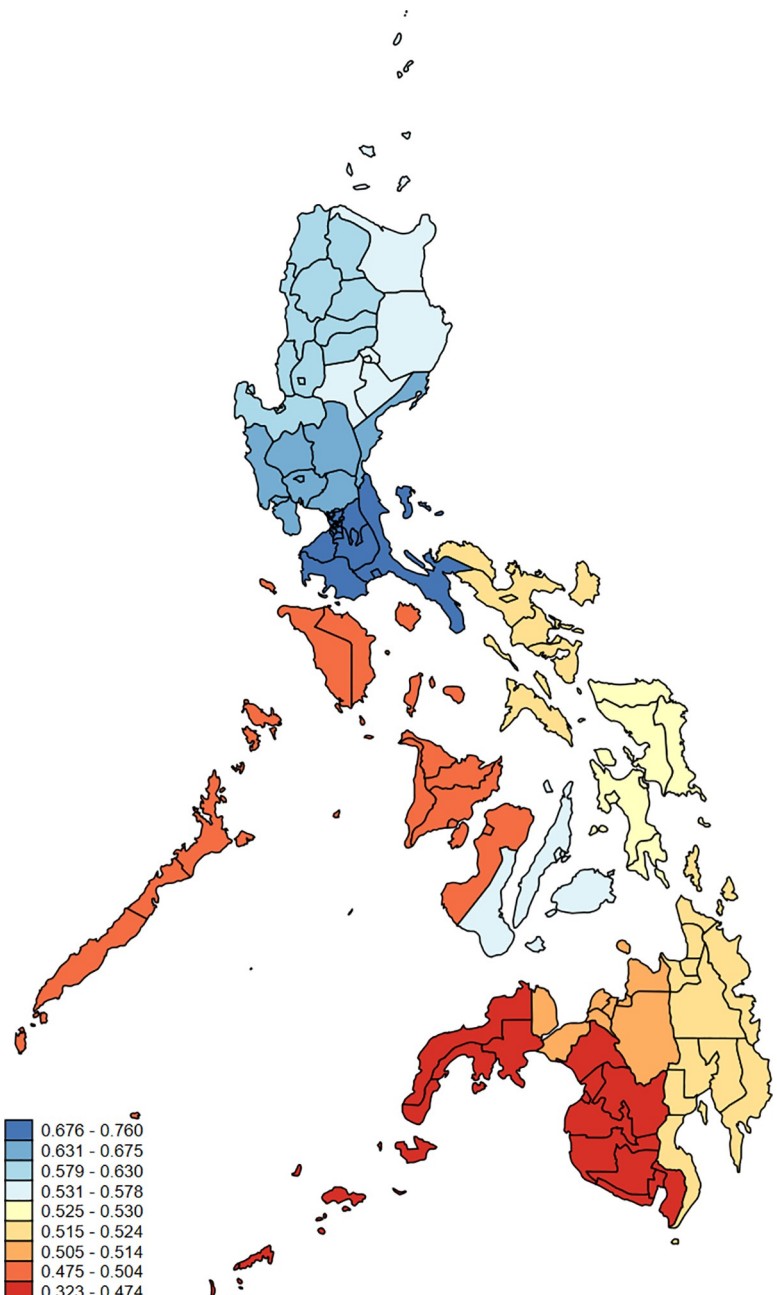

**Fig 2. Mapping regional variation in child-based capabilities.** Fig 2 shows the child-based capability index across first-level administrative units in the Philippines. Source: authors' calculations using child data from the Philippines DHS (2017) and a base map provided by Natural Earth (https://www.naturalearthdata.com/) ($N$ = 22,158). DHS, Demographic and Health Surveys.

birth, and calculated a health index using the estimated life expectancy for each household [13]. Our approach relied on fewer assumptions and is methodologically straightforward to extend to other indicators, populations, and settings [42]. We illustrated it with a limited number of countries, though it can easily be reproduced in other contexts using, for instance, Multiple Indicator Cluster Surveys (MICS) data [43]. Likewise, it can be replicated with alternative outcomes (e.g., child growth failure [44–46]) to examine progress along the development

spectrum with specific prevention programs across and within countries (e.g., nutrition programs, vaccination coverage). Our results can also be mapped for geographical regions within countries (as illustrated in Fig 2), to point decision-makers and public health practitioners to more targeted efforts to improve outcomes among populations in at-risk regions [47,48].

Despite the overwhelming evidence of the associations between the core dimensions of human capabilities, few comprehensive measures presently exist to track investments in all 3 dimensions of the HDI jointly. Our illustrative computation of a child-based capability index —a child-based version of the HDI—relates to a handful of parallel initiatives that have focused on summary metrics of health [49], of education and health [31,50], and of children's well-being [51,52] (see S2 Text for additional details). The World Bank, for instance, introduced a Human Capital Index in 2018, which combines indicators of health and education into a measure of the human capital that a child born today can expect to obtain by her 18th birthday [31]. These summary metrics have been suggested as complements in policy analyses rather than replacements of the HDI (one need not be an alternative to the other) [49]. Few recent efforts, however, have been made to expand the measurement to include education, health, and economic growth—and, to our knowledge, none have looked at indicators that are specifically focused on improving child health at the national and subnational level.

Nevertheless, our study presents a number of limitations. First, this is a descriptive study that explores patterns and trends in human capabilities and child health in LMICs but does not aim to determine causality between components of the child-based capability index. Second, we aggregated child outcomes using first- or second-level administrative units that were available in the DHS surveys. In the future, a more granular look at the HDI may improve the resolution of our findings (e.g., at the village level). Our results for the subnational child-based capability index in India, for instance, may mask substantial heterogeneity within states and union territories. Third, nationally representative household surveys are a relatively expensive and infrequent source of detailed population data [21]. Future research efforts are needed to determine whether alternative approaches are feasible to estimate the different components of the child-based capability index more frequently and economically. Machine learning techniques, for instance, have been recently applied to data from mobile phones, social media, and satellites to estimate demographic and socioeconomic indicators, including population densities [53] and household wealth [54]. Fourth, while our approach for the child-based capability index is relatively straightforward to apply by practitioners universally, the types of policy interventions required to improve child health may vary by country and setting.

In conclusion, this study maps patterns and trends in human capabilities and is among the first, to our knowledge, to introduce a child-based capability index at the national and subnational level. Areas of chronic deprivation may indicate within-country poverty traps and require alternative policy approaches to improving child health in low-resource settings. These findings may point decision-makers working towards achieving the Sustainable Development Goals to more targeted efforts to further reduce persistent health disparities.

## Supporting information

**S1 Checklist. RECORD checklist.**
(DOCX)

**S1 File. Study design.**
(DOCX)

**S1 Text. Measure of household wealth.**
(DOCX)

**S2 Text. Additional details on sensitivity analyses.**
(DOCX)

**S1 Fig. Underlying principles of the child-based capability index.**
(DOCX)

**S2 Fig. Child-based capability index using infant mortality.**
(DOCX)

**S3 Fig. Child-based capability index using full birth history.**
(DOCX)

**S4 Fig. Heterogeneity in child-based capabilities, by sex.**
(DOCX)

**S1 Table. Child-based capability index by subnational region.**
(DOCX)

**S2 Table. Child-based capability index using alternative specifications.**
(DOCX)

**S3 Table. Comparison with other national-level indices.**
(DOCX)

**S4 Table. Comparison with other subnational index.**
(DOCX)

## Author Contributions

**Conceptualization:** Jan-Walter De Neve, Kenneth Harttgen, Stéphane Verguet.

**Data curation:** Jan-Walter De Neve, Kenneth Harttgen.

**Formal analysis:** Jan-Walter De Neve, Kenneth Harttgen, Stéphane Verguet.

**Investigation:** Jan-Walter De Neve, Kenneth Harttgen, Stéphane Verguet.

**Methodology:** Jan-Walter De Neve, Kenneth Harttgen, Stéphane Verguet.

**Project administration:** Jan-Walter De Neve, Kenneth Harttgen, Stéphane Verguet.

**Resources:** Jan-Walter De Neve, Kenneth Harttgen, Stéphane Verguet.

**Software:** Jan-Walter De Neve, Kenneth Harttgen.

**Supervision:** Jan-Walter De Neve, Kenneth Harttgen, Stéphane Verguet.

**Validation:** Jan-Walter De Neve, Kenneth Harttgen, Stéphane Verguet.

**Visualization:** Jan-Walter De Neve, Kenneth Harttgen, Stéphane Verguet.

**Writing – original draft:** Jan-Walter De Neve.

**Writing – review & editing:** Jan-Walter De Neve, Kenneth Harttgen, Stéphane Verguet.

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
