## [Decision Letter · Decision Letter 0]

13 Nov 2019

Dear Dr. De Neve,

Thank you very much for submitting your manuscript "A child-based Human Development Index: estimates from a nationally and regionally representative analysis of 1.7 million under-five children" (PMEDICINE-D-19-02961) for consideration at PLOS Medicine. 

Your paper was discussed among the editorial team and sent to independent reviewers, including a statistical reviewer. The reviews are appended at the bottom of this email and any accompanying reviewer attachments can be seen via the link below:

[LINK]

In light of these reviews, we will not be able to accept the manuscript for publication in the journal in its current form, but we would like to invite you to submit a revised version that fully addresses the reviewers' and editors' comments. You will appreciate that we cannot make a decision about publication until we have seen the revised manuscript and your response, and we expect to seek re-review by one or more of the reviewers. 

We hope to receive your revised manuscript by Dec 04 2019 11:59PM. Please email us (plosmedicine@plos.org) if you have any questions or concerns.

Please let me know if you have any questions. Otherwise, we look forward to receiving your revised manuscript in due course. 

Sincerely,

Richard Turner PhD, for Louise Gaynor-Brook, MBBS PhD

Associate Editor, PLOS Medicine

rturner@plos.org

In the data statement, please make that "Data are ...". We suggest adding "... and all relevant study data are included in the paper" or similar. 

Please restructure your title so that the portion after the colon consists only of the study descriptor, e.g., "...: a cross-sectional study". We suggest: "Nationally and regionally representative analysis of 1.65 million children aged under 5 years using a child-based human development index: a cross-sectional study".

Please combine the "methods" and "findings" subsections of your abstract, and add a new final sentence to the combined subsection to summarize the study's main limitations. 

After the abstract, we ask you to add a new and accessible "author summary" section in non-identical prose. You may find it helpful to consult one or two recent research papers published in PLOS Medicine to get a sense of the preferred style. 

Please trim the paragraph at lines 84-96 to briefly convey the aims of your study, removing mentions of your findings ("we identify ...") and moving elements of discussion to the discussion section. 

Early in the methods section of your main text, please state whether the study had a protocol or prespecified analysis plan, and if so attach the relevant document(s) as a supplementary file (referred to in the methods section). Please highlight analyses that were not prespecified. 

We ask you to restructure the first paragraph of your discussion section so that it consists predominantly of a summary of the study's findings. Aspects of discussion can appear in subsequent paragraphs. 

Throughout the text, please use the past tense consistently to describe analyses, e.g., at line 85 "... we used nationally and regionally ...". 

Please substitute 1.65 million for 1.7 million throughout. 

Please consider possible stigmatizing interpretations of the term "hot spots". 

In your reference list, please abbreviate journal names as appropriate. 

Please add a completed checklist for the most appropriate reporting guideline, which may be RECORD, as a supplementary file, referred to in your methods section. In the checklist, individual items should be referred to by section (e.g., "Methods") and paragraph number rather than by line or page numbers, as the latter generally change in the event of publication. 

Comments from the reviewers:

*** Reviewer #1: 

The paper presents an interesting method of triangulating maternal education and household wealth with a third factor that is health-related, in this case, under-5 child survival. The purposes are to 1) create a new child-based human development index (HDI) at the sub-national (e.g., province) and national levels; and 2) create visual representations in the form of heat maps that show where the highest child mortality is located along regional education and wealth dimensions. The authors provide a sample of countries with data from 2 different decades to show evidence of change (or lack of change) over time using the heat maps.

I think that the greatest value of the author's approach is developing an index that operates at the sub-national level, which can inform country policy makers and program implementers of at-risk regions within the country. At the national level, I'm unconvinced that the child-based HDI will reveal important insights that could not be obtained from ranking or comparing countries using a number of other indices (e.g., the WB's Human Capital Index, the SDI, WPS, or others). Can the authors provide evidence of improved sensitivity to inequalities or needs with their index compared to some of the other popular indices (not just the HDI)? In other words, what would motivate a researcher to choose their index at the national level over others?

I have several requests for additional clarification regarding the indices:

1) Th DHS supplies survey sampling weights to recreate a representative sample of the population. Were these used when calculating the indices? If so, please add this information. If not, shouldn't they be?

2) The authors rely on the DHS sampling frame to select the unit of aggregation for the sub-national index. In some countries, there are as few as 3 regions (e.g., Malawi) and in others, as many as 25 regions (e.g., Peru). Why did the authors not use second level administrative units for the sub-national indices, when available? It seems as though they may have used second level units for some of the heat maps (see below). Why the inconsistency between the admin unit used for the index and the heat maps for a given country (I understand why it might vary by country)? 

3) The authors point out repeatedly that the advantages of their sub-national index over that of Harttgen and Klasen [13] are that they don't use imputation and that child mortality is more sensitive than life expectancy. While I agree that using actual data over imputation is an advantage, the argument (and the paper) would be stronger if the authors made a direct comparison of the 2 methods and demonstrated their sensitivity advantage. 

Regarding the heat maps, more detail is needed to understand how they were created. What software package was used? Was there some kind of smoothing routine? If lower than first level administrative units were used, this should be specified. For example, Malawi has only had 3 regions - or 3 data points for all 3 dimensions. Presumably more than 3 units were used for the heat maps. Is this true?

I also think that the heat maps would be more useful if they were overlaid with geographic regions within countries (i.e., with geospatial hot spot analysis). Take the example of the Philippines, which the authors point out has a persistent hot spot of high child mortality on the wealth and education dimensions. However, we don't know if the hot spot is in fact in the same geographic region by looking at the two heat maps. If we assume that it is, the next questions would be - why that region? Is it particularly isolated and difficult to get services to? Is it in an area of political instability and on-going conflict? It would be helpful if the authors went into more detail about how the heat maps would be used by a policy maker or program planner, and they could use the Philippines as an example.

Other comments

1) I'm confused as to why the authors write that this paper provides a better understanding of how wealth, education, and under-5 survival are "co-produced." Maybe they mean where these factors are co-produced or co-exist? It would be helpful to me (and perhaps other readers) if the authors could provide some clarification of this concept and/or more of a theoretical background for their intent in applying this term here.

2) The authors could be more accurate in describing the locations on the heat maps and avoid using household or individual level terms that imply something that they cannot really say about individual households or mothers. Since the maps reflect aggregates of within-country regions, they cannot, for example, tell us about poor households or uneducated mothers within an overall wealthy region. The text might read something like: low survival shifted from regions with more wealthy households to regions with more poor households.

3) I think that the paper could be tightened by cutting at least one of the long lists of countries, the additional list of administrative units, and avoiding repetition about the details of [13] that uses imputation. This would leave room in the introduction to explaining how the paper supports "co-production" of the 3 human capital components and in the discussion for how the heat maps can be used by policy-makers.

4) In the methods, page 5, the description of the survival indicator needs re-organization. I would recommend moving the information on page 6 about birth history earlier and rephrasing the sentence: "…create a binary variable indicating whether the child was alive or not…" Which child? The wording is confusing without understanding the source of the child information. Also, please add the information about which women are eligible to be interviewed about their birth history in the DHS (i.e., their age).

5) In the analysis section, top of page 9 - second sentence, did the authors mean to write "administrative unit" in the parentheses - not "household"?

6) In the analysis section, how were household wealth and maternal education assigned to individual children before aggregating? Presumably, the same household wealth indicator was assigned to all children in the household, and the maternal education to all children from her birth history record? 

7) The authors went to a great deal of effort to run a number of important sensitivity analyses. I was interested in knowing more about what they learned from these. What can be learned from the fact that the results were relatively insensitive to a wide range of specifications?

8) Figure 1 - Notes - The description of the x and y components is reversed - wealth and education, not education and wealth.

9) In the discussion, the authors present possible explanations for the surprising results from Egypt and Peru. Did they consider stratifying the child survival data by the sex of the child to examine if this finding applied equally to boy and girl children? 

10) I think the authors might note that their sub-national index still masks within administrative unit variation. States in India are as big as some countries with significant within state variation.

*** Reviewer #2 (statistical reviewer): 

This is a useful and well-conducted study on the development of a child-based Human Development Index (HDI). The study design, datasets, and statistical methods and analyses are mostly adequate and of a good standard. The development of the HDI formula is relatively simple and straightforward although it's a bit debatable to use only maternal education as the education indicator. The authors claimed that the proposed HDI provides more detailed heterogeneity in child health within and across countries over time, which seems fine. However, there are still a few issues needing attention.

1) The heatmap is a bit difficult to read and follow. The usability, visualisation and interpretation is very important for an index to be accepted and widely used. An online tool / interactive website for the proposed HDI with all instructions and explanations would be useful to showcase and promote the index.

2) Some form of validation of the index would be helpful. How do we know the proposed index placed among other similar/available HDIs? similarities and differences? A side by side comparison with other indices would be very helpful. Also, although there are 55 countries, can authors please take one country out to comprehensively illustrate and explain the usefulness of the index?

3) Overall, the discussion is a bit too brief. Need to be more comprehensive and critical.

*** Reviewer #3: 

The topic of this paper is interesting. The paper was written well. However, I did not find important implications of the proposed method. First, it requires individual data that not every country can have for every year. Second, how to measure household wealth to be comparable between urban and rural, across regions of a country and among country is challenging. The current method used based on ownership of household assets and quality of the dwelling is controversy. Third, that health is based on only one indicator might not be sensitive enough to detect small differences among countries.

***

[LINK]

---

## [Decision Letter · Decision Letter 1]

23 Jan 2020

Dear Dr. De Neve,

Thank you very much for re-submitting your manuscript "Nationally and regionally representative analysis of 1.66 million children aged under 5 years using a child-based human development index: a cross-sectional study" (PMEDICINE-D-19-02961R1) for review by PLOS Medicine.

I have discussed the paper with my colleagues and the academic editor and it was also seen again by two reviewers. I am pleased to say that provided the remaining editorial and production issues are dealt with we are planning to accept the paper for publication in the journal.

[LINK]

We look forward to receiving the revised manuscript by Jan 30 2020 11:59PM. 

Sincerely,

Louise Gaynor-Brook, MBBS PhD

Associate Editor 

PLOS Medicine

plosmedicine.org

Requests from Editors:

General comments: 

Please round the total number of 1,657,272 children down to 1.65 million in your title and Author Summary. Please use the accurate total number of 1,657,272 children throughout the main text. 

Please remove spaces between refs in square brackets where more than one are cited.

Please revise your title to "Nationally and regionally representative analysis of 1.65 million children aged under 5 years using a child-based human development index: a multi-country cross-sectional study". Apologies for another minor revision to your title. 

Abstract Background - please define what is meant by ‘human capabilities’ 

Abstract Methods and Findings: Please make clear the range of scores that can be generated by the child-based HDI; whether these are normalised

Abstract Conclusions - Line 45: Please add ‘to our knowledge’ or similar to avoid assertions of primacy

Author Summary: Please remove border from Author Summary. 

Please revise the first two bullet points of ‘Why was this study done?’ to use non-identical language to your abstract/introduction.

In the first bullet point of ‘What did the researchers do and find?’, please revise to ‘1.65 million children under five years of age’

In the first bullet point of ‘What do these findings mean?’, please clarify what is meant by ‘child capabilities’ 

In the second bullet point of the same, please revise to ‘These findings may point decision-makers’

Please revise ‘persistent health disparities toward the Sustainable Development Goals.’

In the final bullet point of ‘What do these findings mean?’, please describe the main limitations of your study.

Introduction

Please indicate in your Introduction whether your study is novel and how you determined that. 

Line 64 - please revise to ‘was initially calculated’

Please remove the study results and/or conclusion from the Introduction, and conclude the Introduction with a clear description of the study question or hypothesis. 

Line 90 - please remove sentence beginning ‘We show heat maps of health…’; 

Line 94 - please remove sentence beginning ‘Our approach to compute…’ (more appropriate for Discussion)

Methods

Thank you for providing a completed RECORD statement. Please add the following statement, or similar, early in the Methods section: "This study is reported as per the REporting of studies Conducted using Observational Routinely-collected Data (RECORD) guideline (S1 Checklist)."

Please refer to your prospective protocol / analysis plan early in the Methods section. Please indicate whether any changes (including those made in response to peer review comments) were made to the plan, with rationale.

Please clarify how data on ‘all children born in the past ten years’ relates to children under five years of age, as is the focus of your study. 

Results

Line 268 - please include that results in Figure 1 are only displayed for selected countries

Line 270 - please consider an alternative term for ‘depth’ (of under-five survival)

Line 278 - please revise to ‘on average’

Discussion

Line 350 - Sentence beginning ‘In Colombia, Egypt, and Nigeria…’ does not seem to naturally follow on from previous sentence, outlining that improvements over time do not occur in all countries. Please clarify.

Line 388 - please remove ‘also’

Line 415 - please begin your one-paragraph conclusion with ‘In conclusion’. Please add ‘to our knowledge’ or similar to avoid assertions of primacy.

Line 218 - please revise to ‘These findings may point decision-makers...’

Text S3 - where p values are given, please specify the statistical test used 

Comments from Reviewers:

Reviewer #1: The authors responded very well to previous comments and this paper has much improved as a result of their additional efforts. I only have two minor remaining comments:

The last paragraph in the introduction needs some re-working to clarify the aims and how the gap discussed in the first 2 paragraphs are meant to be filled. For example, it is never explicitly stated that the authors aimed to create an index in this paragraph to address issues with the existing indices. The sentence: "Second, we used under-five mortality, household wealth, and maternal educational attainment at the sub-national level as our measures of health, wealth, and education, respectively" leaves us hanging - they used these measures to do what?

The word "data" is plural. Sentences with the clause "…survey data was available…" should be "…survey data were available…"

Reviewer #2: Thanks authors for their great effort to improve the manuscript. I am mostly satisfied with the response and revision. However, for my 2nd point on validation and comparison with other existing indices, the response was satisfactory but didn't appear in the main text of the paper. The validation and comparison is a vital part of the paper so it should be mentioned and explained explicitly in the results section in the main paper. I can see this was explained in details in supplementary information Text S3 but it should appear or be mentioned in the main paper, as least briefly.

[LINK]

---

## [Editor Report · Decision Letter 2]

10 Feb 2020

Dear Dr De Neve, 

On behalf of my colleagues and the academic editor, Dr. Margaret Kruk, I am delighted to inform you that your manuscript entitled "Nationally and regionally representative analysis of 1.65 million children aged under 5 years using a child-based human development index: a multi-country cross-sectional study" (PMEDICINE-D-19-02961R2) has been accepted for publication in PLOS Medicine. 

PRODUCTION PROCESS

PRESS

PROFILE INFORMATION

Thank you again for submitting the manuscript to PLOS Medicine. We look forward to publishing it. 

Best wishes, 

Louise Gaynor-Brook, MBBS PhD

Associate Editor 

PLOS Medicine

plosmedicine.org